

# Finding novel relationships with integrated gene-gene association network analysis of *Synechocystis sp.* PCC 6803 using species-independent text-mining

Sanna M. Kreula[1,2], Suwisa Kaewphan[2,3,4], Filip Ginter[4] and Patrik R. Jones[5]

[1] Department of Biochemistry, University of Turku, Turku, Finland
[2] University of Turku Graduate School, University of Turku, Turku, Finland
[3] Turku Centre for Computer Science (TUCS), Turku, Finland
[4] Department of Future Technologies, University of Turku, Turku, Finland
[5] Department of Life Sciences, Imperial College London, London, United Kingdom

Corresponding author
Patrik R. Jones,
p.jones@imperial.ac.uk

## ABSTRACT

The increasing move towards open access full-text scientific literature enhances our ability to utilize advanced text-mining methods to construct information-rich networks that no human will be able to grasp simply from 'reading the literature'. The utility of text-mining for well-studied species is obvious though the utility for less studied species, or those with no prior track-record at all, is not clear. Here we present a concept for how advanced text-mining can be used to create information-rich networks even for less well studied species and apply it to generate an open-access gene-gene association network resource for *Synechocystis sp.* PCC 6803, a representative model organism for cyanobacteria and first case-study for the methodology. By merging the text-mining network with networks generated from species-specific experimental data, network integration was used to enhance the accuracy of predicting novel interactions that are biologically relevant. A rule-based algorithm (filter) was constructed in order to automate the search for novel candidate genes with a high degree of likely association to known target genes by (1) ignoring established relationships from the existing literature, as they are already 'known', and (2) demanding multiple independent evidences for every novel and potentially relevant relationship. Using selected case studies, we demonstrate the utility of the network resource and filter to (*i*) discover novel candidate associations between different genes or proteins in the network, and (*ii*) rapidly evaluate the potential role of any one particular gene or protein. The full network is provided as an open-source resource.

## INTRODUCTION

*Synechocystis sp.* PCC 6803 (hereafter *Synechocystis* 6803) was the first photobiological organism to be sequenced in 1996 (*Kaneko et al., 1996*). It is a unicellular prokaryote with a compact genome (~3.5 Mbp) that is capable of non-facilitated DNA-uptake and homologous recombination. It has been extensively studied as a model for photosynthesis

and cyanobacteria in general (*Ikeuchi & Tabata, 2001*), and more recently it has been considered also as a potential host for biotechnology in which solar energy is directly converted into chemical energy and feedstock (*Rosgaard et al., 2012*).

Compared to other photobiological model species, such as *Arabidopsis thaliana* (*De Bodt et al., 2012*), there is still a relative lack of systems biology resources for *Synechocystis* 6803 and cyanobacteria in general. The online 'CyanoBase' portal has played an important role in providing information from genome sequencing data for the cyanobacteria community (*Nakao et al., 2010*). However, as far as we are aware, there are only two other online databases for easy access of a collection of omics data sets (CyanoEXpress (*Hernandez-Prieto & Futschik, 2012*) and RegCyanoDB (*Nair, Chetty & Vinh, 2017*)). Transcriptome data sets included in the CyanoEXpress repository have mainly been analyzed in respective original publications by differential or simple clustering analysis; Efforts to utilize cyanobacteria systems biology data sets for graph-based network analysis are otherwise rare (*Bhadauriya et al., 2007*; *Lv et al., 2015*) Similarly, there is only one online graph-based network analysis platform that includes cyanobacteria species (STRING (*Franceschini et al., 2013*)). The STRING network, however, lacks cyanobacteria-specific data apart from its genome sequence. To complicate matters further, the majority (55.1%) of genes in CyanoBase remain "unknown" (*Fujisawa et al., 2017*). In part this reflects the early date of the first sequencing and persistence of historical archives of annotations in some databases; however, it also reflects the fact that very few studies have been carried out with *Synechocystis* 6803 in comparison with other model species. For example, 350,630 articles including the term '*Escherichia coli*' were found in PubMed July 2017 whilst only 3,853 included the term '*Synechocystis*' (Fig. S1).

Text-mining is a developing technology with increasing potential for scientific utilization, especially given the recent trend towards open access in the scientific literature (*Gonzalez et al., 2016*; *Van Landeghem et al., 2011*). One opportunity with text-mining is to aggregate knowledge from the massive volume of available literature and generate detailed maps of knowledge that would be difficult to obtain otherwise. Naturally, the utility of such network-based aggregation depends on the quantity and quality of the source data (Fig. S1), as well as the method of extracting the information, aggregating it and visualizing it in a meaningful manner for humans. The lack of existing literature for poorly (or not at all) studied organisms is typically addressed by clustering homologous genes into groups (gene families) based on sequence homology (*Van Landeghem et al., 2011*). Relationships between any two gene families can then be extracted from the entire accessible literature, allowing species-independent bibliome networks to be created. This has significant implications for lesser studied species as it considerably broadens the quantity of available data for network construction.

Intuitively, a text-mining network comprises interactions that are already 'known' and thus not 'novel' in the strict sense. Novel interactions can be hypothesized, through indirect connections that involve two or more known connections, but these will lack evidence unless that information was present in the article but not extracted. Therefore, in order to identify novel connections that are more likely to be true, we integrated a PubMed and PubMed Central based bibliome network with complementary networks

created using available large-scale experimental data sets (transcriptome, protein-protein interaction). This increases the network size (i.e., number of nodes and edges) and the presence or absence of complementary omics-based edges lends positive or negative support, respectively, for indirect text-mining links. The criteria for a genuinely interesting novel relationship was then set to require at least two independent pieces of 'evidence'. Hence, in order to facilitate the search for potential novel gene-gene associations in large networks, we developed a filter to identify only those interactions that are (1) not directly linked by text-mining events yet (2) supported by links from multiple data sources. This then allows a search for both novel genes in sub-systems of interest and identification of a context (and thereby possible biological role) for orphan genes aided by gene ontology analysis. This study illustrates that text-mining not only helps identify novel genes with particular physiological, regulatory or metabolic roles but also allows network clusters and patterns with likely coordinated functions to be identified.

We are interested in the metabolism of cyanobacteria, as a potential host for sustainable biotechnology. As a proof of concept, we therefore first applied this methodology to create a network resource for the cyanobacterium *Synechocystis sp.* PCC 6803 and provide case study examples with a focus on metabolic processes of interest, including the metabolism of NADPH, nitrogen, Fe-S and alkanes.

## METHODS

### Construction of the networks

Molecular interaction networks were retrieved and constructed from publicly available databases and from the literature, as follows:

#### *Networks constructed using microarray and yeast-2-hybrid data*

To create a *Synechocystis* 6803 co-expression network, 68 data sets from a large scale transcriptomics study (*Singh et al., 2010*) were used. The transcriptome data was collected and stored as fold change (log2 (treatment/control)) of gene expression values in tab-delimited text files. The data was thereafter subjected to further analyzing after importation into the analyzing and visualizing platforms Cytoscape 2.8.2, 3.0.1 and 3.3.1 (*Shannon et al., 2003*; *Smoot et al., 2011*), depending on available plugins. The ExpressionCorrelation plugin (*Hui et al., 2008*) was employed to generate a co-expression network using the expression values. A similarity matrix was calculated using the Pearson correlation coefficient with a strength threshold of $\pm 0.7$. In order to identify an appropriate threshold, the ExpressionCorrelation plug-in histogram function was used to select a reasonably sized network. The obtained co-expression based gene network (1,886 nodes and 10,187 edges) is referred to as CoEx. A second yeast two-hybrid (abbreviated Y2H) protein-protein interaction network was constructed by importing into Cytoscape a list of identified protein-protein interactions from an available data set (*Fields & Song, 1989*; *Sato et al., 2007*).

#### *Text-mining data*

BioContext (*Gerner et al., 2012*) and EVEX (*Van Landeghem et al., 2013b*) are one of only few resources providing gene/protein association from the entire literature databases that
have been made publicly available. Both EVEX and BioContext apply their text-mining tools to PubMed abstracts and PubMed Central (PMC) full-text articles and are able to extract a similar number of events (36 vs. 40 million for BioContext and EVEX, respectively).

Besides having more extracted events, EVEX also aggregates events of genes in the same family providing hypothetical network where gene/protein association can be conveniently inferred from closely related sequences. Though both resources are considered equivalent extracting the same information from the same data set, we consider EVEX database a more suitable text-mining resource for our studies, due to its data on gene-family association and more recently updated articles.

The network from the EVEX database is composed of two data sets following the different releases of EVEX namely, EVEX-2011 and EVEX-2013. EVEX-2011 (*Van Landeghem et al., 2011*) is the first public release of the EVEX text-mining database which covers the literature up until June 2011 (http://www.evexdb.org/ (*Van Landeghem et al., 2013a*; *Van Landeghem et al., 2013b*; *Van Landeghem et al., 2011*)). EVEX-2013 (*Van Landeghem et al., 2013b*) was released with the extended coverage of articles from June 2011 up to June 2012 and an updated gene family assignment. Both of the EVEX data sets (EVEX-2011 and EVEX-2013) were combined and used in the present study.

EVEX data was generated using natural language processing tools primarily based on machine learning (ML) to automatically extract cellular processes and interactions among genes and their products such as RNAs and proteins (genes for short). The tools perform three significant steps namely "name entity recognition", "event extraction" and "name entity normalization", which will be discussed here briefly. Firstly, the tools perform name entity recognition by identifying the gene mentions in the documents. The systems then extract the biological events for each gene mention by identifying words or phrases discussing cellular process such as *regulation* and *phosphorylation* and link them to corresponding genes. Finally, to be able to link the genes to information in other databases, genes are normalized by mapping to the Entrez Gene database and respective family identifiers. In case of organism ambiguity, i.e., when the organism is not explicitly stated for a particular mention thus preventing it from being normalized to a single unique identifier, the mention is only mapped to a gene family. Full details of the EVEX text-mining pipeline generating has been described previously (*Van Landeghem et al., 2013b*).

The text-mining network was constructed by retrieving *Synechocystis* 6803 genes (nodes) and their associations (edges) from the EVEX database. This was extended by the addition of associations of all *Synechocystis* 6803 gene homologs originating from the Ensembl resource and the GENIA corpus (*Kim, Ohta & Tsujii, 2008*).

The EVEX network was further enriched with additional information obtained from the EVEX database. The edge attributes included all the organisms for which the relationship was reported, the taxonomic distance between each organism and *Synechocystis* 6803, fine-grained details of the relationship such as the types of the regulation (positive, negative and unspecified), speculation, negation and text-mining prediction confidence score. The node attributes also include *Synechocystis* 6803 gene descriptions, symbols, synonyms, and Entrez Gene identifiers.

Finally, in the NCBI Entrez Gene record, the functions of a well-characterized gene are described by human annotators based on experimental evidence. While generally very useful, oftentimes these descriptions lack specificity, e.g., for genes annotated as "hypothetical". Further, new sequences with no supporting evidence naturally lack this annotation altogether.

To address this deficiency, we also provide descriptions based on the single most prevalent function among the genes in each family. For a small gene family (i.e., <5 genes), the diverse descriptions can be manually combined and selected to represent the common functions of genes in a given family. However, this process is not suitable for a large family with thousands of genes. Here we used the method called "canonicalization" described in (*Van Landeghem et al., 2011*) to select the representative description of the family. First, we collected the descriptions of all genes in a family from NCBI Entrez Gene records. We then reduced the orthographic differences by lowering the case and removing all non-alphanumeric characters such as empty space, parentheses and apostrophes. The description of the gene family is the most common canonical form of descriptions shared by most genes in the family.

The three networks, CoEX, Y2H and EVEX, were thereafter integrated using the Cytoscape tool "Advanced network merge". The merge was carried out based on the Entrez Gene identifiers. For those data sets that did not contain such node identifiers, these were obtained by mapping through CyanoBase gene identifiers. The resulting merged network (IntNet) is provided as File S1 and the attribute annotations are listed in File S2. Although the constructed networks inherently contain complex structures (*Kashtan et al., 2004*), the present work has only focused on simpler triad motifs.

## Annotations for genes defined as "unknown" and "hypothetical"

In this study, we were interested in the information gained for non-annotated proteins when integrating multiple types of data. We primarily used annotation data from CyanoBase downloaded on 22nd of June 2012 to annotate the network. Genes which were not annotated or annotated as 'unknown' or 'hypothetical' in CyanoBase were instead annotated with their gene family description from Entrez Gene as described above. The latest CyanoBase annotation for *Synechocystis* 6803 genome sequence released in 2015 is used as gold standard for comparison with gene family assignment. The data was downloaded on 2nd of February 2018.

## Automated rule-based filtration using a script

Guilt-By-Association networks were created by extending selected nodes through existing edges in the network to include also their direct neighbours, an automated process in Cytoscape termed "First neighbours of selected nodes (undirected)". The filter was developed to find simple triangular patterns (three nodes connected by three edges, also called a triad motif (*Milo et al., 2002*)) from the integrated network, in order to identify relationships between selected key genes (i.e., known or relevant genes for the interested study) and candidate genes (potentially related to key gene) that are most likely to be of interest. The rules of the filter were defined as follows, except where indicated: (*i*) The

triangular pattern needs to have at least two different data-types and (*ii*) no direct EVEX edge originating from *Synechocystis* 6803 is allowed between a key-gene and a candidate gene, as it is therefore already known. In order to facilitate future follow-up analysis, the clusters were also ranked. The ranking of the entire pattern was given according to the following order: (1) EVEX (link coming from article based on *Synechocystis* 6803), (2) EVEX (link coming from article based on any other organism than *Synechocystis* 6803), (3) CoEx, (4) Y2H. Additional ranking rules were constructed to classify the most relevant candidate genes; (*j*) does the putative candidate have additional interactions with other key genes, (*jj*) do genes with direct interactions have additional indirect links and (*jjj*) do additional direct or indirect interaction exist in the extracted pattern. These rules prioritize candidates that are well connected within the network and more related to the metabolism involving key genes.

The script for the filter was written in Python to query the integrated network via the Cytoscape plugin CytoscapeRPC (*Bot & Reinders, 2011*). CytoscapeRPC recognizes the script as a client and allows the script to query or modify the networks. The purpose of the filter was to identify candidate genes (CG script) related to known key genes in metabolism of interest. A second script was also prepared in order to allow the functional prediction of "hypothetical protein" (HP script), i.e., by identifying the function of unknown proteins from a group of functional proteins they are associated with. The ranking is identical to the filter script where we only substituted the role of "key genes" and "candidate genes" (File S3) with "functional protein" (i.e., proteins with verified function) and "hypothetical protein" (File S4) respectively.

## Computational requirements and potential applications on other organisms

The *Synechocystis* 6803 network is relatively small compared to other organism networks such as humans which have in general both larger numbers of nodes and edges (e.g., 13,418 nodes and 265,738 edges) (*Hakala et al., 2013*). The time required to generate networks is thus only a matter of seconds on a general desktop machine. However, the integration of the network requires identifier compatibility, a general problem in integrating data from different database sources, e.g., NCBI Entrez Genes and Taxonomy databases. In this study, this task took us a few hours to manually ensure the compatibility and accuracy of the data.

## Text mining performance

Due to the variance and ambiguity inherent to human language, extracting biological knowledge from text is fundamentally a demanding task requiring a complex system composed of multiple components. While most individual components of the systems are typically evaluated in isolation by their respective developers, evaluating the integrated system is difficult due to the relative lack of gold-standard data sets. In our previous work, we estimated the performance of TEES, the text-mining system used in creating the EVEX database, by manually evaluating the text-mining network of *E. coli* NADPH metabolism. The result showed that the system can perform well on event extraction and gene family assignment, achieving 53% and 72% accuracy, respectively (*Kaewphan et al., 2012*). The

two estimates roughly correspond to, and further verify the evaluation results of TEES on human metabolism (*Björne et al., 2010*). Therefore, we can expect the accuracy of the system in the extraction of the *Synechocystis* 6803 network to be similar as well.

## RESULTS AND DISCUSSION

A major challenge in the evaluation of complex biological networks that have not been manually curated is to know if any of its relationship links (i.e., network edges) are (1) novel and (2) correct. By integrating networks built from experimental data and text-mining it should be possible to rapidly tell whether relationships suggested from experimental data are already known *a priori* from the literature or, the reverse. If the underlying analytical data is independent and complementary to the text-mining data, it should also be possible to boost our ability to evaluate the relative likelihood that a relationship in the integrated network is true or not (through cognitive or rule-based interpretation). This assumes that multiple pieces of evidence from genuinely independent experimental data, all implying a similar conclusion, will increase the likelihood that a suggested relationship is true. In the present study, these two concepts were applied to create a meta-network based on two network-types: (1) experimental ((*i*) transcriptome and (*ii*) protein-protein interactome) and (2) literature. The methodology was applied to the metabolism of *Synechocystis* 6803 as a specific case study.

### Network construction

A species-independent text-mining network (here abbreviated EVEX) was created by first assigning all genes in the *Synechocystis* 6803 genome to gene families using Ensembl Genomes (*Kersey et al., 2012*). All events extracted using the TEES software (*Van Landeghem et al., 2013b*) for these selected gene families were thereafter compiled and imported into Cytoscape (*Cline et al., 2007*). The thus created text-mining network was therefore composed of all machine-readable interactions (defined *a priori*, i.e., 'examples of event triggers') between any two gene families that contain at least one homolog in *Synechocystis* 6803, accessing all literature for all species in PubMed abstracts and PubMed Central Open Access full-texts up to June 2012. In this network, the nodes represent *Synechocystis* 6803 gene symbols and edges linking the nodes represent relationships (grouped into categories of binding, regulation or indirect regulation) between gene families. As a comparison, the text-mining network created using publications studying only *Synechocystis* 6803 (85 nodes, 81 edges) was significantly smaller than that using the species-independent approach (806 nodes, 3,023 edges).

For the transcriptome-based network (here abbreviated CoEx), a co-expression network was constructed using a collection of published microarray data that until now only had been collectively studied with a data-degrading normalization using discrete values (*Singh et al., 2010*). We created a co-expression network (1,886 nodes, 10,187 edges) with the Cytoscape plugin ExpressionCorrelation (*Hui et al., 2008*). For the protein-protein interaction network (here abbreviated Y2H), we used an available qualitative protein-protein interaction data set (1920 nodes, 3236 edges) generated in a high-throughput screening with the yeast-two hybrid method (*Fields & Song, 1989*; *Sato et al., 2007*).

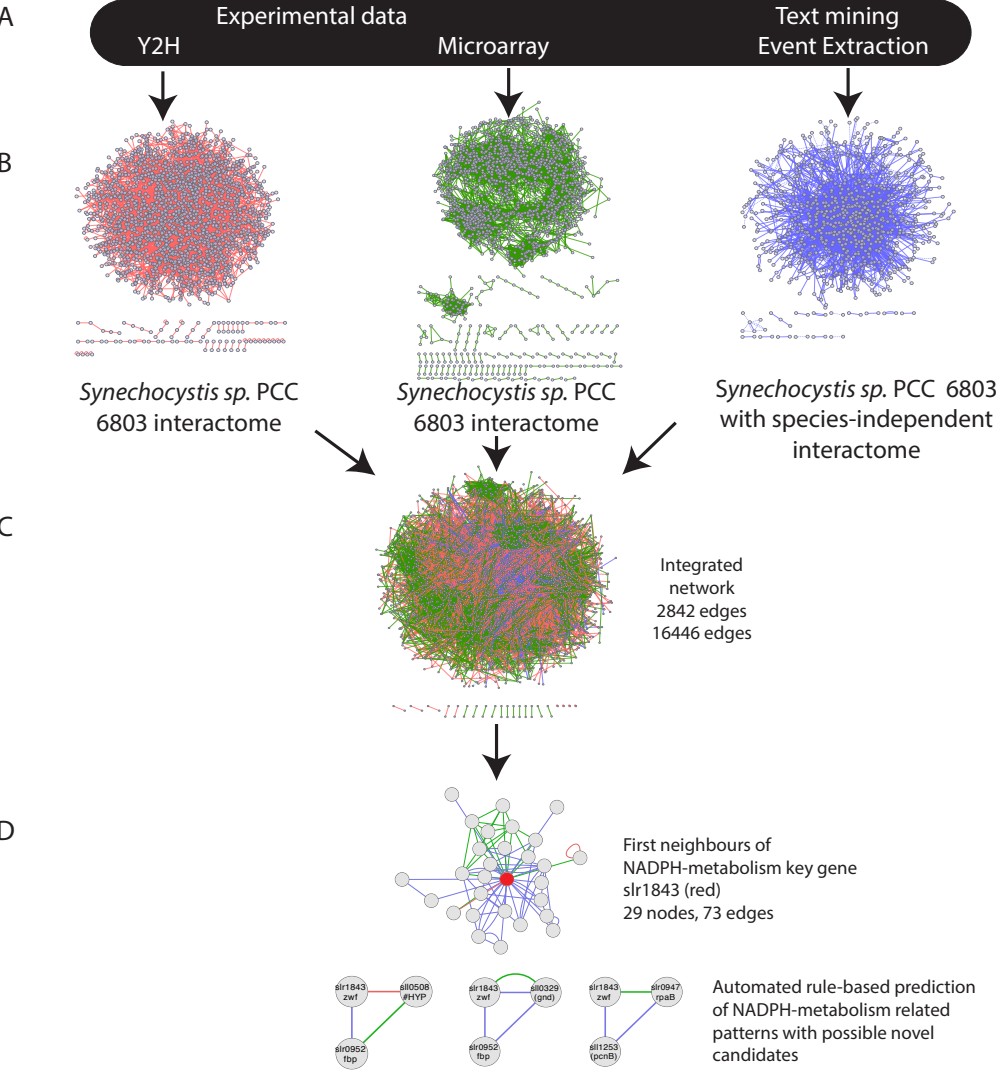

**Figure 1 Overview of the approach—integration of networks created using three distinct data-types.**
(A) The selected data sets Y2H, microarray and text-mining were retrieved and pre-processed. (B) Networks were constructed in Cytoscape and (C) merged (IntNet) with the "advanced network gene slr1843 was extracted by guilt-by-association (GBA). Automated rule-based prediction was used to extract patterns with possible novel candidate genes. A spring embedded layout was used to construct the Cytoscape view. Data-types are visualized with different colours (Y2H, red; CoEx, green; EVEX blue) to easily distinguish between them.

The integration of all three networks in Cytoscape using the advanced network merge plugin resulted in a combined network (IntNet) of 2,842 nodes and 16,446 edges (File S1), representing 76% of the genome and all of its native plasmids (*Kaneko et al., 1996*) (Fig. 1). In order to ensure that all the three integrated networks were independent, two edges in the EVEX network (slll0041-sll0269, sll0041-slr1636), which originated from the paper first reporting the data used for the Y2H network, were removed from IntNet.
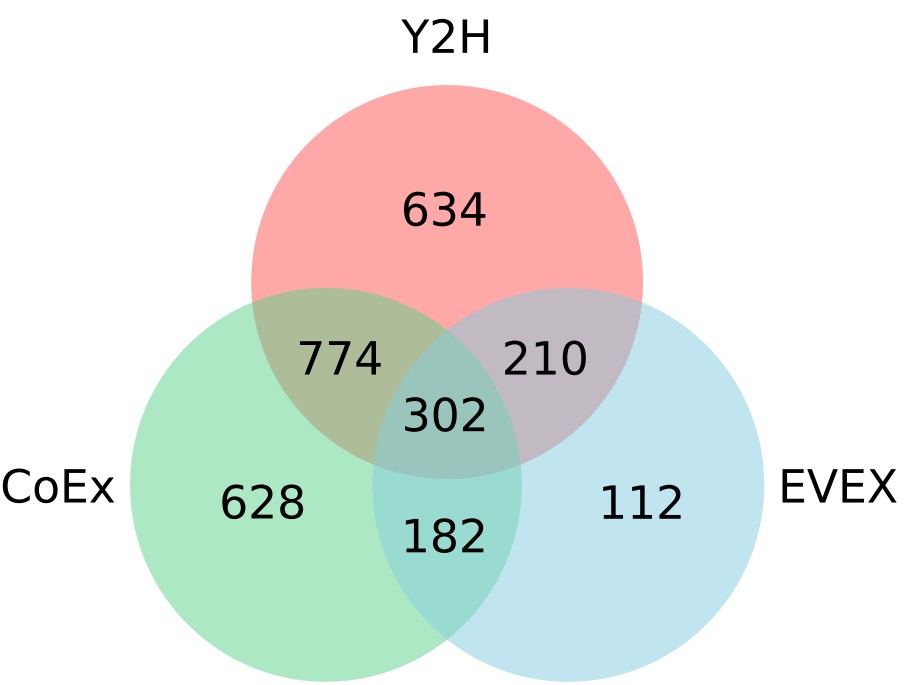

**Figure 2** **The distribution of nodes across the three (Y2H, CoEx and EVEX) networks.**

## Global properties

Overall, IntNet displayed surprisingly little overlap between different data-types. While 52% of the nodes (1,468) are represented in at least two networks, only 11% are represented in all three (Fig. 2). The distribution of source organisms used in the species-independent text-mining network is summarized based on domains and supergroups in Fig. 3. Most relationships in the EVEX network originate from studies with bacteria, the same domain of life as *Synechocystis* 6803. Within the Bacteria domain *Escherichia coli* dominates, reflecting the number of publications in PubMed (Fig. S1). The second most represented group of organisms that contributed to the *Synechocystis* 6803 text-mining network belongs to the Metazoa, with human, rat and mouse being the most common contributors.

An additional benefit with the integration of different data-types was the enhancement in the number of meaningful annotations afforded by combining annotations in CyanoBase (*Nakao et al., 2010*) with those provided by the gene family assignments. In the microarray data set 1913 genes (46.5% of genome) were annotated (from CyanoBase) as 'hypothetical' or 'unknown'. The integration with the species-independent text-mining network increased the number of meaningful annotations in the complete network (IntNet) by 402 additions (from 53.5% to 67.6% of the genome) through the addition of gene family annotations (listed in File S5).

To shed light on the quality of automatic functional annotation using gene family assignment, we used the latest annotations of *Synechocystis* 6803 from Cyanobase database (*Fujisawa et al., 2017*) as gold standard data set to compare with our annotation. The comparison focused on the family assignments which were used on proteins defined as
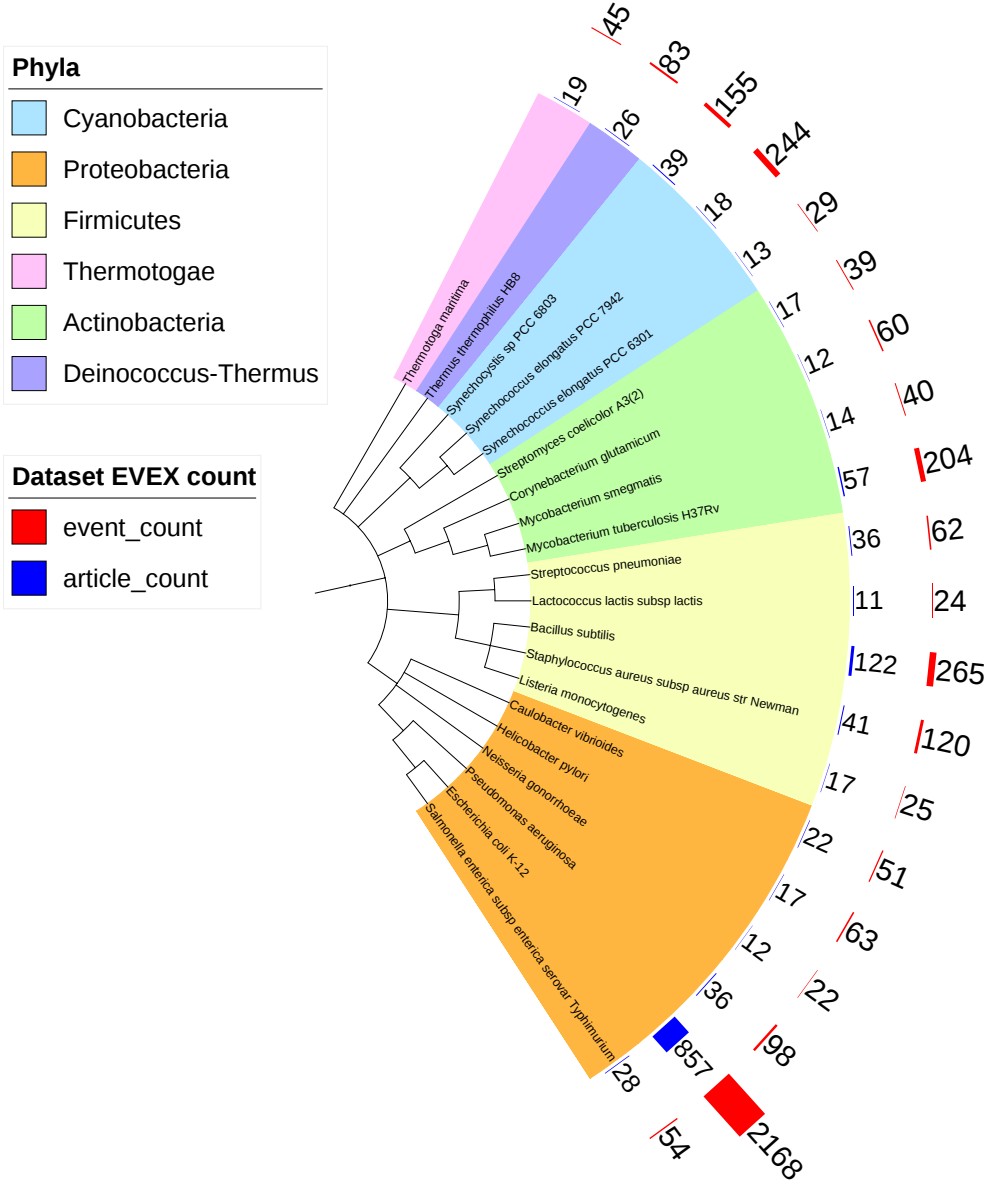

**Figure 3** **The phylogenetic origin of the text-mining events used to construct the species-independent network.** *Escherichia coli* K-12 is the most studied organism as demonstrated by the biggest red (number of events) and blue (number of articles) bars. Only the species (all prokaryotes) that contributed most to the species-independent network are shown.

'unknown' or 'hypothetical' in CyanoBase version 2012, not the difference between the two releases. Among 402 annotated genes, the functions of 126 *Synechocystis* 6803 genes (31.3%) are still unknown or unannotated, while our approach suggests the functions shared by most of the proteins in the family. About one third of our annotations overlapped, with 67 genes (16.7%) receiving an annotation identical to CyanoBase 2015, and 54 genes (13.4%) where either annotation is a substring of the other. The rest of the annotations, 38.6%,
either differed in their specificity, one being more general than the other, or were entirely unrelated. Overall, we can see that functional annotation based on gene families add value over even the latest Cyanobase release, providing annotation to a number of genes which otherwise would have had none, and can thus be used to facilitate the annotation process and support the interpretation of results.

## Automated rule-based filtration of candidates with a high likelihood of real relationship

Smaller first neighbour (Guilt-By-Association, GBA) sub-networks were first constructed for each of the case study key gene sets. Our impression was that although GBA networks were very useful, the associated cognitive interpretation (here defined as 'manual') was dominated/biased by already existing knowledge and/or relationships only supported by a single data type. In addition, it is possible that potentially interesting relationships would not be perceived owing to the daunting complexity of larger GBA networks. We therefore developed a filter to identify smaller motifs (also called clusters) that would enhance the search for potentially novel and relevant relationships between selected key genes (known or relevant genes for the study of interest) and candidate genes (having potential relationship to key gene(s)). The filter was set to demand at least two different data-types between a key gene and candidate gene, and direct EVEX edges between key genes and candidate genes were allowed only if they did not originate from a study using *Synechocystis* 6803. The output from the filter is both different and complementary to a conventional GBA analysis since (1) relationships based only on existing knowledge (i.e., direct EVEX edges) with key genes are discarded, and (2) only patterns with multiple supportive evidence (i.e., more than one edge-type) are accepted. Despite these efforts, an unknown proportion of the edges in IntNet, and motifs extracted therefrom using the filter, are still likely to be false positives.

## Utilization of the integrated network to obtain novel biological insight

What can we use IntNet and its filtered derivative networks for? The diverse utility of interaction networks has been described previously (*Franceschini et al., 2013*). Apart from general properties and patterns on a genome-scale level (as described above) we considered two utilities of particular value for biological studies using lesser studied species: (1) To identify novel candidate genes with potential relationships between two or more key genes representing an important biological process, and (2) to probe the possible role of an otherwise unknown gene or gene set that has been identified by other means. The first utility would be particularly valuable with poorly studied organisms for the collation of members of pathways or other similar systems that do not display co-existence in the form of operons. The second utility, on the other hand, would be important as a follow-up to other studies that have identified genes or proteins by experimental means (e.g., affinity chromatography, yeast-2-hybrid). To evaluate these utilities, we employed key gene sets from selected case studies (Table 1) to (*i*) extract first neighbor GBA-clusters and (*ii*) sub-clusters generated from all candidate genes (and associated triangular patterns) derived using the automated script. The key gene sets were decided prior to the study
**Table 1  List of key genes used in the case studies.** Key genes identified for alkane biosynthesis were based on the consensus operon structure in cyanobacteria (*Klähn et al., 2014*).

|  | Gene name | Annotation (Cyanobase) |
| --- | --- | --- |
| **NADPH metabolism** | | |
| slr1239 | *pntA* | pyridine nucleotide transhydrogenase alpha subunit |
| slr1434 | *pntB* | pyridine nucleotide transhydrogenase beta subunit |
| slr1843 | *zwf* | glucose 6-phosphate dehydrogenase |
| slr1289 | *icdh* | isocitrate dehydrogenase |
| slr1643 | *fnr* (PetH) | ferredoxin-NADP oxidoreductase |
| ssl0020 | *petF* | ferredoxin I |
| **Iron sulfur cluster metabolism** | | |
| sll0088 | *sufR* | hypothetical protein (transcriptional regulator, *suf*) |
| slr0074 | *sufB* | ABC transporter subunit |
| slr0075 | *sufC* | ABC transporter ATP-binding protein |
| slr0076 | *sufD* | hypothetical protein (FeS assembly protein) |
| slr0077 | *sufS/nifS* | cysteine desulfurase |
| slr1417 | *sufA* | hypothetical protein YCF57 (FeS assembly protein) |
| **Alkane biosynthesis** | | |
| sll0209 | *aar* | acyl-ACP reductase |
| sll0208 | *ado* | aldehyde deformylating oxygenase |
| sll0207 | *rfbA* | glucose-1-phosphate thymidylyltransferase |
| sll0728 | *accA* | Acetyl-CoA carboxylase alpha subunit |
| slr0315 | | probable oxidoreductase |
| slr0426 | *folE* | GTP cyclohydrolase I |
| **Sigma factor** | | |
| Sll1689 | *sigE* | group2 RNA polymerase sigma factor SigE |

based on the research interests of the group. The clusters and networks generated by both methods were thereafter evaluated manually in order to verify potentially interesting and novel candidate genes and to benchmark the overall approach.

*Case study 1—novel candidates with a potential relationship to SigE*

SigE (sll1689) is a sigma factor that has been demonstrated to influence central carbon metabolism with broad impact, as evidenced by a shift in the distribution of central carbon metabolites in response to the deletion of *sigE* or over-expression of SigE (*Kloft, Rasch & Forchhammer, 2005*; *Sundaram et al., 1998*). The first neighbour GBA and script-based clusters are shown in Fig. 4A, including several interesting candidates. Firstly, we noted a link between SigE and slr1055 (ChlH), a light- and $Mg^{2+}$-dependent anti-sigma factor shown previously to have specificity for SigE (*Osanai et al., 2009*). However, this link was not based on the article that demonstrated this relationship in the first place (*Osanai et al., 2009*). Instead, SigE connects with ChlH through edges of all three network types, a direct Y2H edge, the lead to identifying the role of slr1055 in the first place (*Osanai et al., 2009*), and indirect edges via sll0306 (SigB, EVEX) and sll1886 (hypothetical protein, CoEx). The experimentally confirmed relationship between ChlH and SigE therefore verifies the
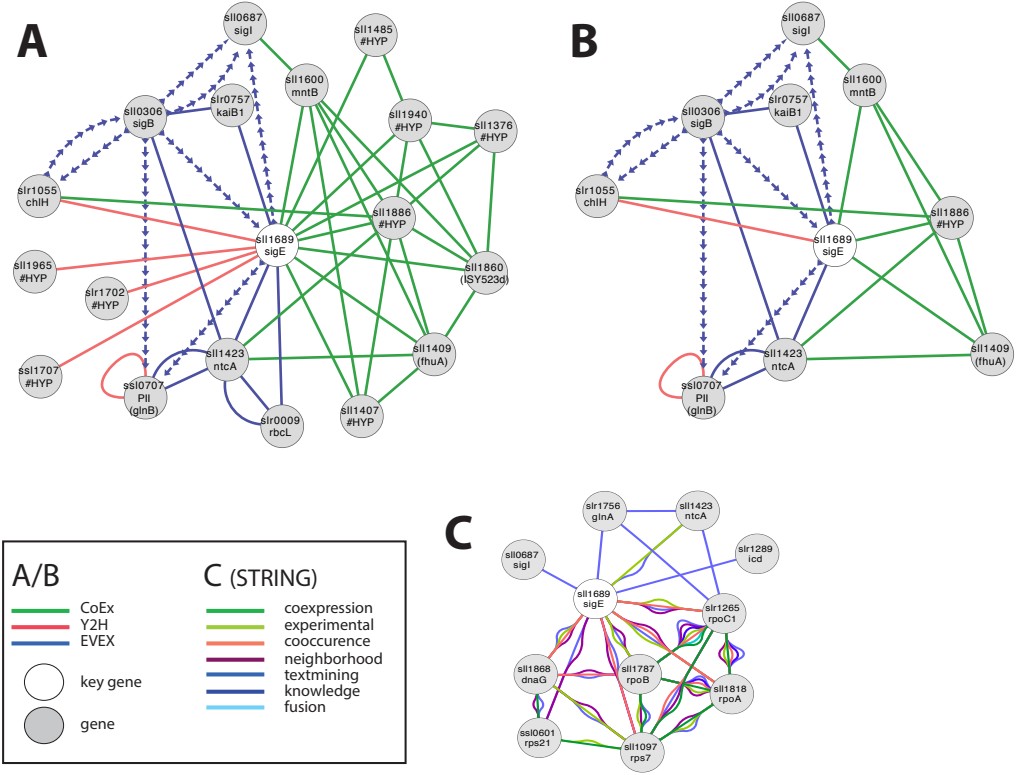

**Figure 4** **The phylogenetic origin of the text-mining events used to construct the species-independent network.** (A) The first neighbor guilt-by-association (GBA) network using only SigE as key gene. (B) The combined network of motifs extracted with the rule-based script. (C) Network generated by STRING database August 23, 2014, using standard settings and sll1689 as input. Solid EVEX edges originate from any organism other than *Synechocystis* 6803. Dotted EVEX edges originate from *Synechocystis* 6803. Black edges originate from STRING database. The key genes are indicated by a white node.

conclusion of the relationship that can be drawn from the present network even in the absence of the direct text-mining link.

Several known proteins with an established role in nitrogen-metabolism (e.g., NtcA, PII (*Kloft, Rasch & Forchhammer, 2005*)), or the circadian clock (KaiB (*Hitomi et al., 2005*)) were also found to be connected to SigE, in addition to others without any meaningful annotation. The automated script (Fig. 4B) suggested a central role for sll1886 (annotated as hypothetical protein) with a close connection to SigE. Sll1886 harbors a putative zinc binding domain and shows weak homology to di-haem cytochrome C (*Vandenberghe et al., 1998*), suggesting the possible involvement of electron-transfer. Interestingly, a manganese transport component (MntB, sll1600) was also part of the script-based cluster which is relevant given that ChlH is $Mg^{2+}$-dependent.

In comparison, we also searched for candidate genes to SigE using STRING-db (*Franceschini et al., 2013*) with sll1689 as input (Fig. 4C). This produced a network of 11 nodes at the default setting. When the script- and STRING-db based networks were compared, the intersection between the two networks contained only three genes; sll1689,

sll1423 (ntcA) and sll0687 (sigI). Interestingly, whilst the STRING network contained an association with glnA, the script-based network contained an association with glnB - both genes have important roles in nitrogen metabolism (*Herrero, Muro-Pastor & Flores, 2001*). Overall, many of the nodes in the STRING network (Fig. 4C) are related to gene transcription (RNA polymerase related gene products), whilst the script-network (Fig. 4B) is dominated by genes with a known role in nitrogen metabolism, as has also been confirmed experimentally (*Muro-Pastor, Herrero & Flores, 2001*). The former network has no 'unknown' members, whilst at least one completely unknown, yet intricately connected, member (sll1886) is present in the latter network. Notably, sll1886 is co-located on the genome to a ''two-component sensor histidine kinase'' (sll1888) which also is a member of the same CoEx network as sll1886 and ntcA (sll1423) (Figs. 4A, 4B). This strengthens the argument that sll1886 may play an important role in nitrogen metabolism.

### Case study 2—NADP(H)-metabolism

The role of the pentose phosphate pathway (PPP) in cyanobacteria under daylight conditions is not entirely clear given that $NADP^+$ is a major electron acceptor of electrons generated by water-splitting photosynthesis. A part of the metabolic flux through the carbon fixing CBB cycle has been measured to pass through the oxidative branch of PPP (oxPPP) under daylight conditions (*Young et al., 2011*) though the optimal solution for biomass flux in stoichiometric models did not incorporate any oxPPP flux (*Knoop et al., 2013*). We were curious about the metabolic role that key-enzymes responsible for $NADP^+$-reduction in fermentative microorganisms may have in an autotrophic system and how they are regulated. The objective in the following analysis was therefore to use the network analysis in order to identify novel candidate genes.

A first neighbour GBA of IntNet with all pre-defined six NADPH key genes generated a complex network of 72 nodes and 194 edges (Fig. 5A) (File S6), including OpcA, the unique cyanobacterial Zwf activator (*Hagen & Meeks, 2001*). In contrast, only two of the 6 key genes listed for NADP(H)-metabolism were retained by the script (Fig. 5B, 18 nodes and 50 edges): Zwf (slr1834, catalyzing the first committed step of metabolic flux into PPP) and Icd (slr1289), catalyzing the only $NADP^+$-reducing step of the TCA-''cycle''.

Looking closer at the script-based network, Zwf forms a motif with slr0952 (annotated as fructose-1,6-bisphosphatase (FBPase)) and sll0508 (annotated 'unknown protein') via three different data-types (Fig. 5C). Sll0508 has low similarity to other proteins and there are no hits from a search with the SIB Motif Scan (incl. Pfam, PROSITE, HAMAP etc.). This slr0952-containing motif is interesting as it suggests a link between oxPPP and gluconeogenesis. In other cyanobacteria, multiple FBPases have been identified and some of the encoding genes are co-located with *zwf* (*Summers et al., 1995*).

Another interesting candidate gene, found only in the CoEx network, is Slr1194. This node is annotated as a '1 protein' that exhibits a high percentage similarity to a 'Mo-dependent nitrogenase family' protein in *Cyanothece sp.* PCC 7424, and links to Zwf via slr1793 (talB) and slr1734. The latter gene is a homolog of OpcA, an allosteric regulator and activation factor of Zwf in other cyanobacteria (*Hagen & Meeks, 2001*).

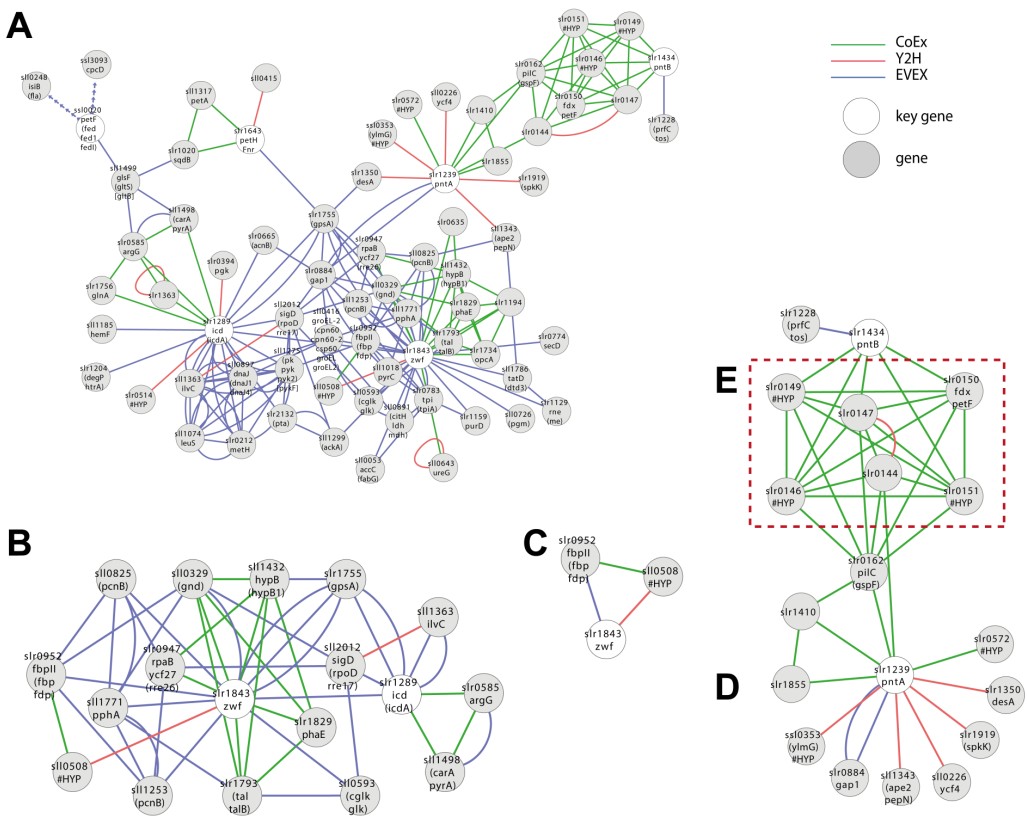

**Figure 5  Cluster analysis with SigE (sll1689).** (A) The first neighbor guilt-by-association (GBA) network using all NADPH-related key genes (Table 1). (B) The combined network of motifs extracted with the rule-based script. (C) Predicted pattern extracted from the script result B. (D) First neighbor GBA using PntA (slr1239) or PntB (slr1434) as input. (E) Red dotted box indicates members of the Pap operon. Solid EVEX edges originate from any organism other than *Synechocystis* 6803. Dotted EVEX edges originate from *Synechocystis* 6803.

Zwf also forms several motifs with *rpaB* (slr0947) that also include the PPP genes *gnd* (sll0329) and *talB* (slr1793) (Fig. 5B). RpaB is a regulator involved in controlling energy transfer between phycobilisomes and PSII or PSI. The relationship between RpaB and genes encoding enzymes in PPP suggests the possibility that also PPP flux may be controlled at least in part by RpaB in response to light quality and/or quantity, or another signal reflecting the internal redox-status.

### Case study 3—Probing the role of an incompletely known gene or gene set—PntAB

*Synechocystis* 6803 harbors two genes (slr1239 (*pntA*) and slr1434 (*pntB*)) encoding a putative dimeric NADPH:NADH-transhydrogenase. PntAB has been shown to catalyze the proton gradient dependent transfer of electrons from NADH to NADP(H) in *E. coli* (*Sauer et al., 2004*). In *Synechocystis* 6803, we would expect under optimal photosynthetic conditions that NADP$^+$ is efficiently reduced by PetH, the Ferredoxin:NADP-oxidoreductase linked to PSI. PntAB may therefore only be important for the supply

of NADP(H) under conditions of limiting light (e.g., during the night) and/or in order to re-oxidize NADH formed by NAD(H)-dependent reactions (*Kämäräinen et al., 2017*). Hence, although PntAB is well-known in fermentative microorganisms it remains unclear what role it may have in cyanobacteria, thereby falling into the category of incompletely known genes.

No motifs satisfying the criteria of the script-based filter were found including either PntA (slr1239) or PntB (slr1434). Nevertheless, a GBA-cluster was extracted using both genes as key genes (Fig. 5D). Both slr1239 and slr1434 form a co-expression based cluster with an operon (slr0144-slr0152) called Pap (Photosystem II assembly proteins) (*Wegener et al., 2008*) and the essential ferredoxin PetF (slr0150; Fig. 5E). The connection is quite convincing as PntA shows CoEx edges with slr0144 whilst both PntB and PetF share CoEX edges with several of the other genes in the operon, though not slr0144. The presence or absence of the Pap operon does not influence growth under so far tested conditions, although deletion mutants display a reduced capacity to evolve di-oxygen (*Wegener et al., 2008*). Why would there be a connection between the Pap operon and PntAB? PntAB has the role in fermentative microorganisms of catalyzing electron-transfer between one major electron acceptor-donor and another, though not ferredoxin. Several genes of the Pap operon are predicted to contain Fe-S clusters, co-factors that typically also are involved in electron transfer, the only common theme so far; this connection deserves further experimental attention to resolve.

### Case study 4—iron sulphur cluster metabolism

As mentioned above, iron-sulphur (Fe-S) clusters are inorganic protein co-factors that are typically involved in electron transfer. They are assembled in cyanobacteria using the SUF system, even though genes with homology to members of the ISC system (the dominant system in *E. coli*) also are present in the *Synechocystis* 6803 genome (*Balasubramanian et al., 2006*). It has been established that SufR (sll0088) is an Fe-S containing negative transcriptional regulator of the remaining *SUF* members (*sufA*, *sufB*, *sufC*, *sufD*, *sufS*) (*Wang et al., 2004*). Interestingly, a first neighbour GBA with all of the above key genes (Fig. 6A, File S6) resulted in a single cluster with two divided parts, an upper part containing all the catalytic SUF members, and a second lower part containing SufR. Even though SufR is clearly the transcriptional regulator of the other SUF members, there is surprisingly no direct connection between SufR and the other SUF members. Instead, SufR forms an intense CoEx cluster with a series of genes annotated mainly as 'hypothetical'. Three of these are iron-related proteins: PerR (slr1738), sll1202 (homolog to iron transporters) and BfrA (sll1341; bacterioferritin homolog). In contrast, the upper SUF operon cluster contains four genes encoding predicted Fe-S containing proteins: The PSI subunit *psaA* (slr1834), *bioB* (slr1364), *sll0031* (hypothetical) and *spoT* (slr1325). A possible reason for the lack of a direct association between SufR and the remaining SUF operon may be that SufR is not the only regulatory factor controlling SUF expression, or that its control is conditional.

Filtration of IntNet using all Fe-S key genes generated two smaller clusters (Fig. 6B). Whilst no obvious insight was obtained from the SufR-containing motif, the second cluster

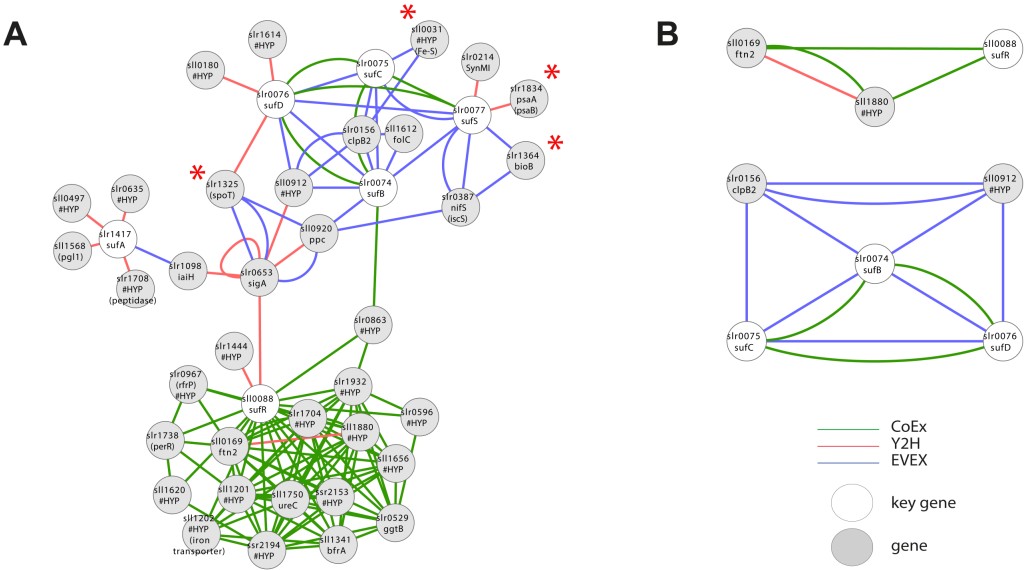

**Figure 6** **Cluster analysis with Iron Sulfur cluster related key genes.** (A) The first neighbor guilt-by-association (GBA) using all members of the SUF operon as key genes (Table 1). Red asterisks indicate genes encoding proteins with a predicted Fe-S cluster binding motif. (B) Two motifs generated by the rule-based filtering script using the same key genes. Solid EVEX edges originate from any organism other than *Synechocystis* 6803. Dotted EVEX edges originate from *Synechocystis* 6803.

contained three SUF operon members connected both by EVEX and CoEx. Interestingly, all text-mining edges originated from a diverse collection of bacteria that did not include any cyanobacteria.

### Case study 5—alkane biosynthesis

The two genes encoding the catalytic enzymes of the alkane biosynthesis pathway (*Schirmer et al., 2010*), and which is present in most but not all cyanobacteria, forms an extended apparent operon in most species where it is found (*Klähn et al., 2014*). Since the alkane biosynthesis reaction so far does not work as efficiently as needed for economically sustainable fuel production (*Eser et al., 2011*; *Kallio et al., 2014*), we were curious whether missing elements required for effective catalysis could be represented in this apparent operon. In *Synechocystis* 6803, however, only three of the apparent operon members are co-located on the genome, sll0207-sll0209. For the assembly of key genes, we therefore included homologs in *Synechocystis* 6803 to the most commonly observed members of the alkane biosynthesis operon in cyanobacteria in general (Table 1), even if they are not co-located on the genome in *Synechocystis* 6803. In this analysis (Fig. 7, File S6), however, most of the operon members did not form a joint cluster with the exception of slr0426. A possible contributing reason for this outcome is that the biosynthetic system is unique to cyanobacteria (*Schirmer et al., 2010*) and that it has not yet been studied much. Consequently, it is not well-represented in the EVEX network.

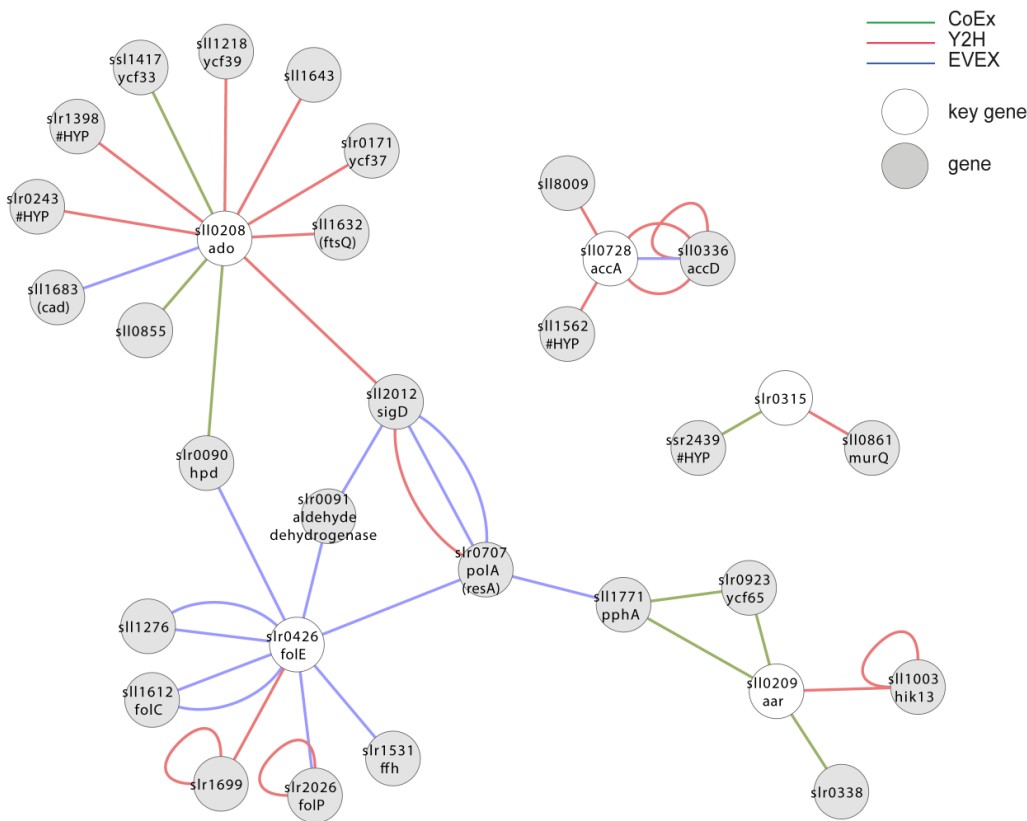

**Figure 7** **Cluster analysis with members of the apparent alkane operon.** The first neighbor guilt-by-association (GBA) of IntNet using two genes encoding catalytic enzymes in alkane biosynthesis pathways and its four most commonly observed co-locating genes in all cyanobacteria. Solid EVEX edges originate from any organism other than *Synechocystis* 6803. Dotted EVEX edges originate from *Synechocystis* 6803. The key genes are indicated by white nodes.

### Case study 6—Screening for the role of genes annotated as 'hypothetical' or 'unknown'

We considered the possibility to utilize the script in order to obtain an insight into the possible role of all genes that are annotated as 'hypothetical' or 'unknown'. The rationale was that the local context of genes without an annotation may provide insight into its possible role and that the script would allow the most important local context to be identified. All genes without an annotation were therefore employed, one at a time, as an entry gene for the automated script. The criteria of this script demanded as previously that more than one relationship type was present, plus the additional new demand that at least one of the members of the local context had an existing annotation. Over 5% of hypothetical/ unknown genes (112/1913) satisfied these criteria. The combined network with filtered motifs was composed of 331 nodes (Fig. 8A; File S7). Around 60% of these patterns were derived from a combination of CoEx/Y2H and around 40% from EVEX/(CoEx/Y2H). These 112 putative genes represent a list of potentially interesting genes to be studied further (File S8). Many of the entry genes with highest ranking have a local context with

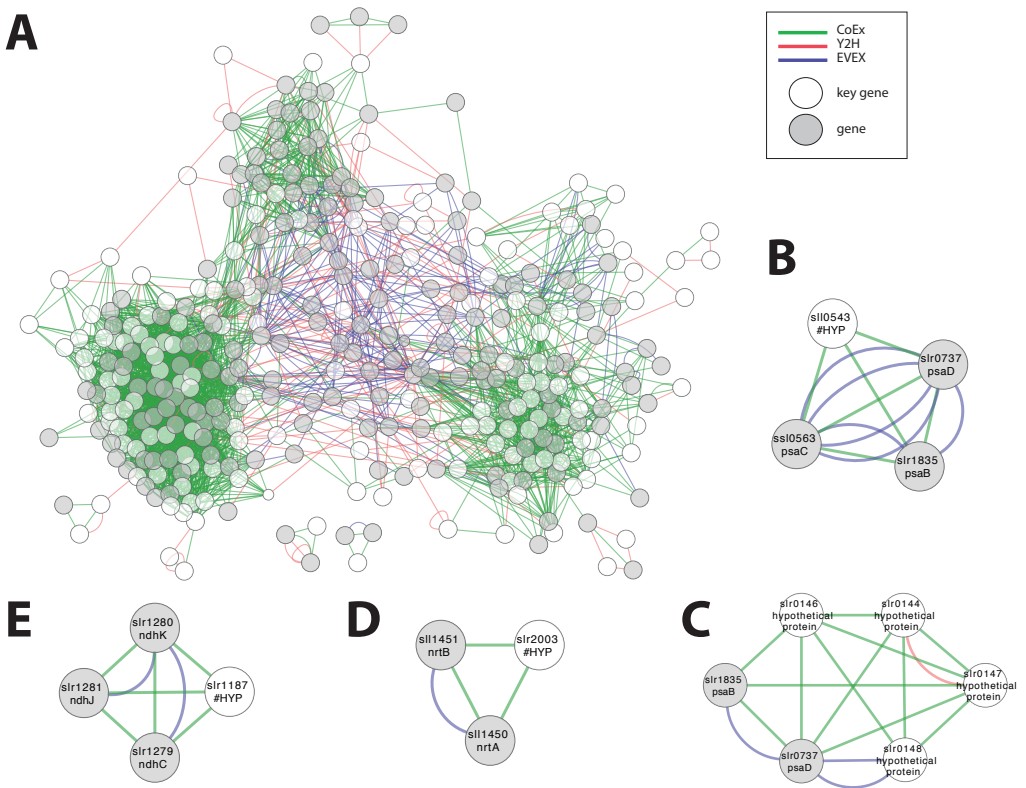

**Figure 8** **Cluster analysis to find the role of genes annotated as 'hypothetical' or 'unknown'.** (A) The combined network of motifs extracted with the rule-based script. (B) sll0543, as an example pattern with highest ranking, forms a cluster with genes encoding three key members of PSI (*psaC*, *psaB*, *psaD*). (C) slr0144-48 as another example (see Fig. 6D). (D) The 'unknown protein' slr1187 forms a cluster with three NADH dehydrogenase subunits (slr1279-81); (E) 'hypothetical protein' slr2003 forms a cluster with two nitrate/nitrite transport system components (slr1450-51).

a clear single focus. For example, sll0543 forms a cluster with genes encoding three key members of PSI (*psaC*, *psaB*, *psaD*) (Fig. 8B). In contrast, a similar analysis with STRING places sll0543 in a cluster of eight genes annotated as 'hypothetical protein' and one as 'indole-3-glycerol-phosphate synthase'. In another example, the slr0144-48 Pap operon (see case study 3) is once again identified (Fig. 8C). Interestingly, in this search, the Pap operon genes form a cluster together with two PSI subunits (psaB and psaD): the only earlier study linked the Pap operon to PSII, not PSI (*Wegener et al., 2008*). Other selected findings include unknown genes slr0723 and sll1774 forming an intricate cluster with two genes encoding proteins with a role in pili biogenesis (slr0161, slr0163) and another gene linked to chemotaxis (slr1043). The 'unknown protein' slr1187 forms a cluster with three NADH dehydrogenase subunits (slr1279-81) (Fig. 8D), and the 'hypothetical protein' slr2003 forms a cluster with two nitrate/nitrite transport system components (slr1450-51) (Fig. 8E).

## CONCLUSIONS

This study incorporates species-independent text-mining for the creation and evaluation of biological networks. Although it is evaluated first with an established model organism, this approach is likely to have even greater utility with "new" species that until now have not been studied, particularly if it can be complemented by omics analysis at a sufficient depth to enable supporting networks to be constructed and integrated with the text-mining network.

Although the analysis of the *Synechocystis* 6803 network was constrained in scope, this hypothesis-generating process uncovered many leads and potential insights into its metabolism and potentially also cyanobacteria in general. For example, the strong apparent connection between the Pap operon and both PSI and PntAB, in addition to PSII as earlier reported. The lack of a clear connection between the alkane biosynthesis genes and other members of its apparent operon in other cyanobacteria was also surprising, though negative. Other leads included sll1886, SigE and nitrogen metabolism, sll0508/slr0952 and NADPH-metabolism, RpaB/slr0947 and PPP, sll0543 and *psaBCD*, slr1187 and *ndhCJK*, and slr2003 and *nrtAB*. Thus, a large number of candidate genes with potential involvement in important biological processes in cyanobacteria were identified in only the small selection of case studies presented here, the entire network certainly contains many more.

The filter allows the potentially most important candidates to be selected given that it relies exclusively on connections that are supported by multiple and independent evidence. It must be pointed out, however, that these automated procedures cannot replace the need for further in-depth cognitive analysis of existing literature, though it may have an important guiding role, and final experimental verification. The script is expected to speed up the identification of the most interesting candidates and allow researchers to place a focus for further cognitive and experimental work, and in so doing contribute to reducing the proportion of 'unknown' or 'hypothetical' proteins.

The analysis of *Synechocystis* 6803 is likely to be further enhanced by future high-quality omics data sets, ideally from the same condition(s). In general, an extension of the EVEX event capture to include also metabolites would enable metabolic stoichiometric networks to also be included. Greater access to full-text articles is also likely to enhance the network richness and accumulation of multiple independent lines of evidence.

## ACKNOWLEDGEMENTS

Computational resources were provided by CSC—IT Center for Science Ltd, Espoo, Finland. We would like to thank Sofie Van Landeghem (Ghent University) and Tero Aittokallio (FIMM) for their insight and valuable comments on text-mining, systems and network biology. In remembrance of the greatest supervisor in my life, Irma Peltonen (S Kreula).

### Funding

This work was supported by Emil Aaltosen säätiö (Sanna Kreula), Academy of Finland projects no. 253269 (Patrik Jones) and no. 138796 (Filip Ginter), ATT Tieto kayttöön grant (Suwisa Kaewphan), and the Finnish Cultural Foundation, Satakunta Regional fund (Sanna Kreula). The funders had no role in study design, data collection and analysis, decision to publish, or preparation of the manuscript.

### Grant Disclosures

The following grant information was disclosed by the authors:
Emil Aaltosen säätiö.
Academy of Finland projects: 253269, 138796.
ATT Tieto kayttöön grant (Suwisa Kaewphan).
Finnish Cultural Foundation, Satakunta Regional fund.

### Competing Interests

The authors declare there are no competing interests.

### Author Contributions

- Sanna M. Kreula and Suwisa Kaewphan conceived and designed the experiments, performed the experiments, analyzed the data, contributed reagents/materials/analysis tools, prepared figures and/or tables, authored or reviewed drafts of the paper, approved the final draft.
- Filip Ginter conceived and designed the experiments, analyzed the data, authored or reviewed drafts of the paper, approved the final draft.
- Patrik R. Jones conceived and designed the experiments, analyzed the data, prepared figures and/or tables, authored or reviewed drafts of the paper, approved the final draft.

### Data Availability

The raw data are provided in the Supplemental File.

### Supplemental Information

Supplemental information for this article can be found online at http://dx.doi.org/10.7717/peerj.4806#supplemental-information.

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
