# Peer review of "Finding novel relationships with integrated gene-gene association network analysis of Synechocystis sp. PCC 6803 using species-independent text-mining"

_PeerJ, doi:10.7717/peerj.4806_

## Round 0.1 · original submission · Major Revisions

Dear authors

After due consideration of the comments offered by 2 referees, I inform you that your manuscript might be accepted for publication after some revisions. As suggested by Rev#3, you should emphasize the the potential of your approach as an hypothesis generator.
Please reply carefully to each comment in detail upon re-submission.
I look forward to receiving the revised version of your manuscript.

Best regards,
Hugo Sarmento

·

Basic reporting

The paper presents a novel and advanced proof-of-concept text-mining pipeline to develop information-rich gene-gene association network for model organisms. The proposed method to improve the accuracy of predicting genetic interactions and identifying novel interactions based on species-specific experimental data integrated with text-mining networks is exciting and useful at the same time. However, there are several key concerns which need careful attention from the authors.

Experimental design

The rationale and advantages behind identifying relationships in a triad pattern could have been elaborated and presented in a better way, because in addition to triad motifs, other motifs have been shown to be used in the literature [1, 2].


1. Kashtan N, Itzkovitz S, Milo R, Alon U: Topological generalizations of network motifs. Physical review E, Statistical, nonlinear, and soft matter physics 2004, 70(3 Pt 1):031909.
2. Milo R, Shen-Orr S, Itzkovitz S, Kashtan N, Chklovskii D, Alon U: Network motifs: simple building blocks of complex networks. Science 2002, 298(5594):824-827.

Validity of the findings

The ranking of the identified patterns seems straightforward; however, it will enhance the inclusion of possible incorrect motifs, and thereby increasing the number of false positives. It seems that being more conservative in accepting the multiple sources of existing knowledgebase is critical. Would the authors include quantitative information on the accuracy of these ranked patterns?

Additional comments

1. The authors should invest further care in rectifying a number of spelling issues throughout the manuscript (for example, page 12, line 271 reads “…Synechocysis 6803…”.

2. Labels (A) and (B) are missing from figure 2.

Reviewer 2 ·

Basic reporting

In addition to Cyanobase, CyanoEXpress and STRING the following references are relevant to the current work

A database on regulatory interactions in cyanobacteria (RegCyanoDB, corresponding article on bioRxiv): https://www.che.iitb.ac.in/grn/RegCyanoDB/

A recent protein-protein interaction network in cyanobacteria: Lv Qi et al. Scientific Reports volume 5, Article number: 15519 (2015). https://www.nature.com/articles/srep15519

L76-77: This sentence is not clear. How is the network expanded? What does “network depth” refer to? The likelihood of making correct predictions (i.e. accuracy) decreases … because of decrease in accuracy.

L80: Please point out in the introduction (and possibly the abstract) that the text-mining is performed on articles in PubMed Central. Can you provide an estimate of how large a fraction this is of all available (relevant) literature?

L282: Using abbreviations (GBA) in figure captions makes life more difficult for the reader. I suggest using replacing GBA by “guilt-by-association” in subsequent figures.

L320: Define abbreviations when they first appear. Using “key gene” and “candidate genes” makes the text more readable than using KG and CGs.

L324-343: This appears to be mostly repetition from the methods section.

L351: “Utility 1” -> The first case.

L500: “Uniquely present in some but not all“ ???

L598: Some of the references are missing/incomplete. E.g. Hui et al. is missing and Kaewphan et al. 2012 and Hakala Kai et al. 2013 are incomplete. Please check the reference section carefully.

Figure 1: This figure does not provide much additional information to what is already stated in the text, in terms of number of publications, E. coli > A. Thaliana > cyanobacteria > Synechocystis. The figure is somewhat trivial and no harm would be done if it were to be omitted.

Figure 2: The caption “species-independent text-mining …” states the obvious. When comparing panels A and B it is not clear that “the same layout was used in both cases”. With ~800 nodes and ~3000 edges the graphical representation of the EVEX network in panel A is not informative. Most of the edges are buried behind the cluster of nodes in the center, which means that the “types” of most connections cannot be inferred from the figure. Omitting this figure would do no harm. The figure in panel B would be more informative if gene IDs were included (the network is probably small enough for this to be reasonable).

Figure 3: It might be worthwhile to highlight the slr1843 node.

Figure 5: The size of the blue circles indicates that the article count is the same for almost all the species other than E. coli. A scale for the event and article counts should be added.

Figure 6: Preserving the network layout between panels A and B would help with comparison.

Figure 7 (and others): What do the loops indicate?

Experimental design

Methods section
* * *
L135-177: The construction of the “EVEX” network is difficult to follow. Was a list of genes from Synechocystis plus a list of genes from organisms from the same family (derived used as input to EVEX which then returns a network (list of nodes and edges)? Maybe some of the text in L254-261 should be incorporated here.

L185: Cyanobase had a significant update in Nov 2016, including annotation updates. What is the reason for using a Cyanobase version from 2012 in this work?

L109: How was the correlation threshold of +/- 0.7 chosen?

L114: “EVEX” refers to a database, a pipeline and a network in the manuscript. Please clarify.

L114: Please motivate the selection of EVEX for the text-mining task. What about using e.g. BioContext (Gerner et al. Bioinformatics. 2012 28(16):2154-61)? You might want take a look at the text-mining effort of Westergard et al. (bioRxiv Jul 11, 2017. http://dx.doi.org/10.1101/162099) which includes articles from Springer and Elsevier as well as PubMed.

L149: The folP and MiaB example is not very helpful.

L155: The network*s*?

L181: Giving the merged network an identifier here (INTNET?) would make it easier to refer to it later on.

L189: The rule-based algorithm is referred to as a rule-based script, rule and pattern-based script, rule-based filtering scripts, automated script, … which is confusing. Please stick to a single term. What about referring to the procedure simplify as a filter?

L190: Please define the “guilt-by-association”. The description of how GBA networks are constructed is confusing. How do you extend a network by adding neighbors to the nodes in the network (where do the neighbors come from?)

L193: Please explain why the triad motif is relevant in this study (three different data sources?) Would it be meaningful to use other motifs? How would you expect the results to change if other motifs were to be used?

L207: “The script” becomes two scripts in L211-213.

L237: Since the number of articles on E. coli in PubMed is approx. 100 times greater than the number of articles on Synechocystis (L54-55) extrapolating accuracy figures from a previous E. coli study to the present Synechocystis study is at best optimistic.

Validity of the findings

L316: Please provide a comparison with the most recent version of Cyanobase (or if that is the case, please update the reference to Fujisawa et al. 2017).

Since there is no experimental validation of the predictions, the use of the method as a "hypothesis generator" should be highlighted.

Additional comments

The authors describe a method for discovering gene-gene relationships by combining experimental data with literature mining. The method may turn out to be useful for studying relatively unknown species, providing an alternative to simple homology search.

The authors provide several case studies demonstrating the use of their methods but without independent (experimental) verification it is not possible to assess the accuracy of the predictions. Case study 6 is particularly interesting since it demonstrates how the method can be used to study hypothetical or unknown genes.

Some parts of the manuscript are difficult to follow and terminology needs to be clarified. Examples and suggestions on how to improve them are given.

---

## Round 0.2 · Minor Revisions

Both reviewers have approved this new version of your manuscript.
There are only a couple of additional minor corrections:

Fig1 (prev. Fig 2): The only thing that the reader can conclude from this figure is that the number of nodes and edges is large (as stated in the text). I suggest omitting it or find a better representation.

L90 I suggest using "network size" instead of "network depth".

L142-143 That BioContext "uses more tools" does not seem relevant.

·

Basic reporting

No further comments

Experimental design

The reviewer's comment has been addressed.

Validity of the findings

The reviewer's comment has been addressed.

Additional comments

The reviewer's comments have been addressed.

Reviewer 2 ·

Basic reporting

.

Experimental design

.

Validity of the findings

.

Additional comments

The authors have addressed all my previous comments in a satisfactory way and I can now recommend the manuscript for publication.

---

## Round 0.3 · accepted · Accept

Your manuscript can now be accepted. Please follow production instructions.

#